# Lactoferrin Induces the Synthesis of Vitamin B6 and Protects HUVEC Functions by Activating PDXP and the PI3K/AKT/ERK1/2 Pathway

**DOI:** 10.3390/ijms20030587

**Published:** 2019-01-30

**Authors:** Huiying Li, Yizhen Wang, Huaigu Yang, Li Liu, Jiaqi Wang, Nan Zheng

**Affiliations:** 1State Key Laboratory of Animal Nutrition, Institute of Animal Science, Chinese Academy of Agricultural Sciences, Beijing 100193, China; thufit2012@126.com (H.L.); m13613613572_2@163.com (Y.W.); yanghgxms@163.com (H.Y.); liliu0324@126.com (L.L.); wangjiaqi@caas.cn (J.W.); 2Key Laboratory of Quality & Safety Control for Milk and Dairy Products of Ministry of Agriculture and Rural Affairs, Institute of Animal Science, Chinese Academy of Agricultural Sciences, Beijing 100193, China; 3Laboratory of Quality and Safety Risk Assessment for Dairy Products of Ministry of Agriculture and Rural Affairs, Institute of Animal Sciences, Chinese Academy of Agricultural Sciences, Beijing 100193, China

**Keywords:** lactoferrin, vitamin B6, TXA2, PGI2, PDXP, HUVEC

## Abstract

As a nutritional active protein in foods, multiple studies of the biological activities of lactoferrin had been undertaken, including antioxidant, antiviral, anti-inflammatory, antitumor, antibiosis, and antiparasitic effects, while the mechanism related with its protection of cardiovascular system remained elusive. In the present work, the effect of lactoferrin on the viability of HUVECs (human umbilical vein endothelial cells) was detected to select the proper doses. Moreover, transcriptomics detection and data analysis were performed to screen out the special genes and the related pathways. Meanwhile, the regulation of lactoferrin in the functional factors thromboxane A_2_ (TXA_2_) and prostacyclin (PGI_2_) was detected. Then, the small interfering RNA (SiRNA) fragment of the selected gene pyridoxal phosphatase (PDXP) was transfected into HUVECs to validate its role in protecting HUVECs function. Results showed that lactoferrin inhibited the expression of TXA2 and activated expression of PGI_2_, as well as activated expression of PDXP, which significantly up-regulated the synthesis of vitamin B6 (VB6) and the phosphoinositide 3-kinase (PI3K)/ serine/threonine-protein kinase (AKT)/ extracellular regulated protein kinases (ERK) 1/2 pathway. For the first time, we revealed that lactoferrin could induce the synthesis of VB6 and protect HUVECs function through activating PDXP gene and the related pathway.

## 1. Introduction

Lactoferrin (80 kDa) belongs to the iron-binding protein family. It contains more than 700 amino-acid residues and exists in several biological secretions, including tears, saliva, and milk, and in the secondary granules of granulocytes [1,2]. Lactoferrin is divided into three types according to the degree of iron (Fe) saturation: the holo-type (with two iron atoms), single iron type (with one iron atom), and apo-type (with no iron atom). The expression levels of lactoferrin in human milk are much higher than in other mammals, and its concentration in human milk decreases over the months of lactation [1,2]. In recent years, multiple studies of the biological activities of lactoferrin have shown that it has many beneficial effects on living organisms, including antioxidant, antiviral, anti-inflammatory, antitumor, antibiosis, and antiparasitic effects [2,3,4,5]. Lactoferrin is also reported to strengthen the immune system, protect the cardiovascular system, reduce gastrointestinal stimulation, improve sleep quality, affect the metabolism of elemental Fe, and balance the concentration of Fe in an organism [6,7,8,9,10]. In recent research, the lactoferrin-based vehicles for the delivery of bioactive compounds were studied and applied, including complexes, emulsions, and nanoparticle parts, indicating that lactoferrin would be a novel source in the field of biomaterials development [11].

Lactoferrin has several physiological properties that are related to cardiovascular health, including the regulation of blood pressure, lipid reduction, antithrombotic effect, etc. [12,13,14,15,16,17]. Supporting its protective effect on the cardiovascular system, lactoferrin has been shown to inhibit angiotensin I-converting enzyme (ACE) activity in human umbilical vein endothelial cells (HUVECs), which may further alleviate the damage caused by oxidative factors, including nitrous oxide (NO), thromboxane A_2_ (TXA_2_), and so on [18,19,20]. However, the particular mechanism by which lactoferrin regulates the expressions of TXA_2_ and prostacyclin (PGI_2_) is rarely reported, and the target of lactoferrin in HUVECs is still unclear. To evaluate the role of lactoferrin in protecting cardiovascular system and regulating TXA_2_ and PGI_2_, as well as to investigate the related mechanism, we carried out a series of experiments. In the present study, HUVECs were treated with different doses of lactoferrin to construct an in vitro model and select the proper dosage; then, transcriptomics detection of cells was performed to screen the specific gene (PDXP), which might play a key role in regulating the functions of HUVECs. Considering the PDXP gene regulates the expression of vitamin B6 (VB6), and TXA_2_/PGI_2_ are regarded as the clinical parameters in evaluating cardiovascular health, the expression levels of VB6, TXA_2_, and PGI_2_ in both the cells and the culture medium were thus determined to investigate the effects of lactoferrin on the model. The transcriptome of these cells was also analyzed to identify the genes specifically involved in these responses, and the roles of these genes in regulating the PI3K/AKT/ERK1/2 pathway were measured with small interfering RNAs (siRNAs) and western blotting.

## 2. Results

### 2.1. High-Dose Lactoferrin Shows No Obvious Inhibition on HUVEC Viability

To determine the effects of lactoferrin on HUVEC viability, different doses (0, 0.05, 0.1, 0.25, 0.5, 1, 2.5, 5, 7.5, 10, or 20 g/L) of lactoferrin were applied, and the cell viability was tested with a Cell Counting Kit-8 (CCK8). There was no obvious reduction in cell viability in the lactoferrin-treated groups (*p* > 0.05) (Figure 1). The cell viability at doses of 5 g/L and 10 g/L were 94.5 ± 4.16% and 94.8 ± 5.67%, respectively, which did not differ significantly from that of the control group (96.7 ± 1.34%). We also confirmed the effect of lactoferrin on HT29 cell viability observed in other published articles, and at doses > 5 g/L, lactoferrin significantly inhibited HT29 cell viability [2], although 5 g/L and 10 g/L lactoferrin did not inhibit HUVEC viability, suggesting that HT29 cells might be more sensitive to lactoferrin. Therefore, these two doses (5 g/L and 10 g/L) were selected as appropriate for the subsequent experiments.

### 2.2. Transcriptomic Analysis of HUVECs and Data Analysis

Transcriptomic analyses were performed in HUVECs before and after treatment with 5 or 10 g/L lactoferrin to identify the important genes in the affected signaling pathway. Three biological replicates were included in each group (control group, 5 g/L group (L), and 10 g/L group (H), each *n* = 3). Based on the p-value (< 0.05) and the absolute fold change (> 2.0), the expression levels of 75 genes and 19 genes were significantly changed, respectively, in the low dosage group and high dosage group (Appendix A), when compared to control group, in which 12 genes were overlapped (Venn diagram in Figure 2A). Among these 12 genes, 10 were up-regulated and 2 down-regulated compared with the control, and pyridoxal phosphatase (PDXP) was the most significant one with a 6-fold change in these 12 genes (Figure 2B,C; Appendix A), this gene being selected to investigate the related mechanism in the present study.

### 2.3. Effects of Lactoferrin and VB6 on the Expression of Thromboxane A2 (TXA_2_) and Prostacyclin (PGI_2_) in HUVECs with Enzyme-Linked Immunosorbent Assay (ELISA) Kits

Lactoferrin clearly inhibited the expression of TXA_2_ and significantly upregulated the expression of vatimin 6 (VB6) and PGI_2_. VB6 (10 μM) significantly reduced the expression of TXA_2_ and increased the level of PGI_2_ compared with the control levels (*p* < 0.05; Figure 3A–C).

### 2.4. Effects of Lactoferrin and PDXP siRNA on the Expression of PI3K, AKT, and ERK1 and Their Phosphorylated Forms in HUVECs

To analyze the effects of lactoferrin and PDXP siRNA in regulating the levels of PI3K, phosphorylated PI3K (p-PI3K), AKT, p-AKT, ERK1/2, and p-ERK1/2 in HUVECs, the levels of these proteins were measured with western blotting. The levels of p-PI3K, p-AKT, and p-ERK1/2 increased after treatment with lactoferrin, whereas the levels of PI3K, AKT, and ERK1/2 decreased significantly in the PDXP-siRNA-treated group compared with the control (*p* < 0.05; Figure 4A,B). However, the three proteins showed no obvious upregulation in the lactoferrin + PDXP siRNA group compared with the PDXP siRNA group (*p* < 0.05; Figure 4A,B).

### 2.5. Effects of Lactoferrin and PDXP siRNA on the Expressions of VB6, TXA_2_, and PGI_2_ in HUVECs

To confirm the regulation of VB6, TXA_2_, and PGI_2_ by lactoferrin and PDXP siRNA in both the cell cytosol and culture medium, the three factors were measured after treatment with lactoferrin, PDXP siRNA, or lactoferrin + PDXP siRNA. Lactoferrin reduced the TXA_2_ levels and increased the PGI_2_ and VB6 levels in both the cytosol and culture medium compared with those in the control group (*p* < 0.05; Figure 5A–C). In the PDXP-siRNA-treated group, TXA_2_ was upregulated, and PGI_2_ and VB6 were significantly downregulated compared with their levels in the control group (*p* < 0.05; Figure 5A–C). However, the levels of these three factors did not differ significantly between the PDXP siRNA group and the lactoferrin + PDXP siRNA group (*p* < 0.05; Figure 5A–C).

## 3. Discussion

Vitamin B6 (VB6) is a small molecule necessary in animal metabolism, occurring as pyridoxal (PL), pyridoxine (PN), pyridoxamine (PM), or their phosphates (PLP, PNP, and PMP) [21,22]. As the main form of VB6, phosphopyridoxal (PLP) is a key coenzyme in 140 kinds of catalyses and participates in several metabolite responses, especially those involving proteins and amino acids [22,23]. PDXP (pyridoxal phosphatase) is composed of 296 amino acids, with a calculated molecular wight (M.W.) of 31.698 kDa, and regulates the dephosphorylation of PLP, PNP, and PMP, after which it transfers the dissociated VB6 (PL, PN, or PM) to the jejunum. The absorption priority of the three substrates is PLP > PNP > PMP [21,22,24]. The metabolic mechanism of PDXP may be related to its dephosphorylation at Ser-3, or with the dephosphorylation of the phosphoserine residue in destrin, which may regulate the recombination of actin during cell mitosis [24].

Gudmundson et al. investigated the protective effects of VB6, in the form of PLP, on the integrity and normal functions of the vascular endothelium in cultured HUVECs incubated with various concentrations of PLP. They found that the PGI_2_ and TXA_2_ produced in HUVECs (detected with radioimmunoassay) can be used as valid indicators of endothelial function [25]. Mahfouz et al. confirmed that VB6 reduces the levels of the superoxide radical and lipid peroxide induced by H_2_O_2_ in vascular endothelial cells, suggesting that VB6 protects HUVECs from oxidative damage [26]. Francesco et al. also showed that VB6 prevents hair loss associated with cancer therapy, which is caused by the serious adverse effects of chemotherapeutic agents on vascular endothelial cells, especially oxidative injury [27]. Thus, VB6 has been shown to play key roles in alleviating oxidative damage and protecting the functions of vascular endothelial cells. However, the mechanisms of these activities require further investigation.

As two antagonistic metabolites of arachidonic acid, TXA_2_ and PGI_2_ belong to the prostaglandin family and are expressed in vascular endothelial cells. A relative balance exists between their expression, which maintains the proper vascular tone and the smooth-vessel blood circulation in the cardiovascular system. Once this balance is disturbed, several diseases can occur, including atherosclerosis, hypertension, cerebral apoplexy, etc. [28,29,30]. In previous studies, endothelial cell injury has been regarded as the early-stage pathological condition underlying atherosclerosis, affecting multiple functions of the vascular endothelial cells, including their permeability, adhesion, kinetics, and proliferation. It also triggers the hyperactive or excessive agglutination of platelets, ultimately resulting in atherosclerosis [31]. The active form of VB6 protects endothelial cells from injury by active platelets, mainly by inhibiting the expression of TXA_2_ and promoting the expression of PGI_2_ [28,29]. Therefore, TXA_2_ and PGI_2_ were analyzed in the present study to evaluate their activities and functions in HUVECs.

In several studies, the PI3K/AKT/ERK1/2 pathway has been shown to induce endothelial cell activation, proliferation, and migration [32,33,34], and this pathway has conflicting biological effects in organisms. On the one hand, the PI3K/AKT/ERK1/2 pathway acts as a positive feedback signal in protecting vascular endothelial cells from endogenous damage because it can promote their proliferation and migration. Meanwhile, PDXP SiRNA intervention was carried out to decrease the expression of PDXP and to investigate the role of PDXP in regulating PI3K/AKT/ERK1/2 pathway. As demonstrated in this study, both the level of VB6 and the expressions of p-PI3K/p-Akt/p-ERK1/2 proteins were down-regulated with the treatment of PDXP SiRNA; thus, we suggests that there might be possible relationship between the synthesis of VB6 and the activity of the PI3K/AKT/ERK1/2 pathway. On the other hand, the PI3K/AKT/ERK1/2 pathway can induce harmful effects, triggering the overproliferation of tumor cells and inducing inflammatory reactions. In our recent research, we demonstrated that lactoferrin inhibited the growth and metastasis of HT29 tumors in a nude mouse model, mainly by suppressing the expression of a vascular endothelial growth factor A (VEGF-A)-related pathway, and PI3K, AKT, and ERK1/2 were shown to be downstream sensors of VEGF-A [2]. However, lactoferrin clearly activates the expression of PI3K, AKT, and ERK1/2 in HUVECs, suggesting that it is a dual regulatory molecule, inhibiting tumor cells but activating normal endothelial cells. This proposition warrants further experimental investigation, which refers to the other mechanistic researches of lactoferrin in protecting cardiovascular system.

Several other genes in transcriptomics detection, besides PDXP (with the most significant 6-fold), were screened out as candidates, and their related mechanistic pathways were also investigated, which will be validated in other researches later. Additionally, the genes related with TXA_2_, PGI_2_, and VB6 were not detected in this research; however, we will pay more attention to them in other upcoming studies.

In conclusion, we investigated the role of lactoferrin in regulating VB6, TXA_2_, and PGI_2_ in HUVECs for the first time and identified the special regulatory role of PDXP with a transcriptomic analysis. We also demonstrated that lactoferrin could regulate the synthesis of VB6 and downstream factors TXA_2_ and PGI_2_ through activating PDXP and the PI3K/AKT/ERK1/2 pathway. VB6 participates in the protection of vascular endothelial cells from oxidative damage, and lactoferrin is an upstream mediator of VB6 expression; thus, lactoferrin has potential to protect the cardiovascular system. Therefore, clinical research into lactoferrin and its medical utility in cardiovascular and cerebrovascular fields is not only important but necessary, and the mechanisms involved require specific attentions.

## 4. Materials and Methods

### 4.1. Chemicals

Lactoferrin, of 95% purity, was purchased from Sigma (St. Louis, MO, USA). The HUVEC cell line was purchased from the American Type Culture Collection (ATCC, Manassas, VA, USA). EGM-2 medium and fetal bovine serum (FBS) were obtained from Gibco (Waltham, MA, USA); 1% streptomycin/penicillin was purchased from Thermo Fisher (Waltham, MA, USA). Cell Counting Kit-8 (CCK8) was purchased from Solarbio (Beijing, China). ELISA kits were purchased from Jiancheng (Nanjing, China). The PDXP siRNA fragment was synthesized by Sangon (Shanghai, China). The primary and secondary antibodies were purchased from Santa Cruz Biotechnology (Santa Cruz, CA, USA). Protein lysis buffer and the related reagents were purchased from Solarbio. Enhanced chemiluminescence (ECL) reagent was purchased from Tanon (Shanghai, China).

### 4.2. Cell Culture and Viability Testing

HUVECs were cultured in EGM-2 medium containing 10% FBS and 1% penicillin/streptomycin in the presence of 5% CO_2_ in a humidified incubator (Thermo Fisher Scientific, Waltham, MA, USA) at 37 °C.

Cell viability was detected and analyzed by CCK 8 assay. About 5 × 10^4^ cells per well were planted into a 96-well plate and incubated for 24 h. Different concentrations of lactoferrin (the stock solution was 20 g/L, with EGM-2 medium as the solvent) ranging from 0-20 g/L in a total volume of 100 μL were then added into the wells and cocultured for 36 h. Then, the medium was removed and 100 μL/well CCK 8 solution was added into the wells for 3 h. The optical density (OD) value was measured by a microplate reader (Thermo Fisher) at the wavelength of 570 nm. The proper dosages for the further experiments were chosen.

### 4.3. Transcriptomic and Data Analyses

HUVECs were cultured and treated with lactoferrin (5 or 10 g/L) for 36 h. They were then collected and treated with TRIzol Reagent (Invitrogen, Carlsbad, CA, USA) to extract the total RNA, according to the manufacturer’s protocol, which was quantified with a NanoDrop 2000 spectrophotometer (Thermo Fisher). The TruSeq Stranded mRNA Library Prep Kit (Illumina, San Diego, CA, USA) was used for mRNA isolation, cDNA transcription, end repair, A-tailing, and adapter ligation. Amplification was performed with 15 cycles and the quality of the products was examined with the Bioanalyzer 2100 System (Agilent, Santa Clara, CA, USA) and a Qubit™ 3 Fluorometer (Thermo Fisher). The libraries were then sequenced on the Illumina HiSeq platform (https://www.illumina.com/) [34], and the data were exported to Excel spreadsheets with Simca-P for a principle components analysis (PCA), partial least squares discriminant analysis (PLS-DA), *t*-test, and variable importance in projection (VIP) plot analysis [34]. The differentially expressed genes between control and lactoferrin-treated groups were identified by the following criteria: (1) *p*-value < 0.05 (*t*-test); (2) absolute fold change > 2.0.

### 4.4. siRNA Treatment of Cells

From the results of our cell transcriptomic analysis and a literature survey, PDXP was selected as a potential target of lactoferrin in HUVECs. A PDXP siRNA fragment was synthesized to investigate the interaction between lactoferrin and PDXP, and to confirm the role of PDXP in regulating the expression of TXA_2_, PGI_2_, and VB6, and the PI3K/AKT/ERK1/2 pathway. HUVECs were plated in a six-well plate for 24 h to 40% confluence. The DNA–liposome complex was prepared and the cells were treated as reported previously [35].

### 4.5. ELISA Detection of TXA_2_, PGI_2_, and VB6

Aliquots (75 µL) of the prepared standard samples were added to the wells containing HUVECs (intensity of about 85%) in a 96-well plate and incubated at 25 °C for 1.5 h. The solution was aspirated from the wells, and the plate was then washed three times with a squirt wash bottle or an automated 96-well plate washer (5 min each). The diluted detection antibody (100 µL) was added to the wells and incubated at 25 °C for 1 h, and then the solution was thoroughly aspirated from the wells and discarded. The plate was washed four times (5 min each). The diluted horseradish-peroxidase-conjugated secondary antibody (100 µL) was added to each well and incubated at 25 °C for 0.5 h. The solution was aspirated, and the plate was washed four times (5 min each). Chromogenic substrate (100 µL) was added to the wells and incubated at 25 °C in the dark for 0.5 h, and then 100 µL of stop solution was added to each well. The plate was evaluated immediately after the solution changed from blue to yellow. Finally, the absorbance of each well at 450 nm and 550 nm was measured, and the OD_550_ values were subtracted from the OD_450_ values to correct for optical imperfections in the microplate (Bio-Rad, Hercules, CA, USA). A curve-fitting statistical software (https://www.ncss.com/software/ncss/curve-fitting-in-ncss/) was used to plot the four-parameter logistic curves for the standards, and the results for the test samples were then calculated.

### 4.6. Western Blotting Detection of Proteins

The cells were collected and the total proteins extracted with lysis buffer containing protease inhibitors. The samples were then centrifuged (4 °C, 10,000 × *g*) for 8 min. After heat treatment at 98 °C, the samples were loaded onto a 10% sodium dodecyl sulfate (SDS)-polyacrylamide gel for electrophoresis and then transferred onto a nitrocellulose filter with a Trans-blot apparatus (BioRad, Hercules, CA, USA). The filter was blocked with 2.5% bovine serum albumin dissolved in phosphate-buffered saline with Tween 20 (PBST) for 1.5 h at room temperature. The proteins of interest were directly probed with the primary antibodies overnight at 4 °C, including antibodies directed against β-actin, PDXP, PI3K, p-PI3K, AKT, p-AKT, ERK1/2, and p-ERK1/2. β-actin was used as the internal reference to ensure equal protein loading. After the filter was washed four time with phosphate buffered saline-tween 20 (PBST) (5 min each), it was incubated with the secondary antibody for 1.5 h at room temperature and then washed four times (7 min each). Finally, the proteins were detected with an ECL reagent (Tanon, Shanghai, China) and analyzed with the ImageJ software (Rawak Software, Inc., Stuttgart, Germany).

### 4.7. Statistical Analysis

All the data are presented as means ± SD. The data were analyzed with the GraphPad Prism 6.0 software (GraphPad, San Diego, CA, USA). All statistical analyses were conducted with Student’s *t*-test or one-way analysis of variance (ANOVA). In the tests of cell viability, qPCR, and western blotting, *p*-values of < 0.05 were considered to indicate statistically significant differences between the control and other groups (* *p* < 0.05), and between the lactoferrin group and the PDXP siRNA group or lactoferrin + PDXP siRNA group (^#^
*p* < 0.05).

## Figures and Tables

**Figure 1 ijms-20-00587-f001:**
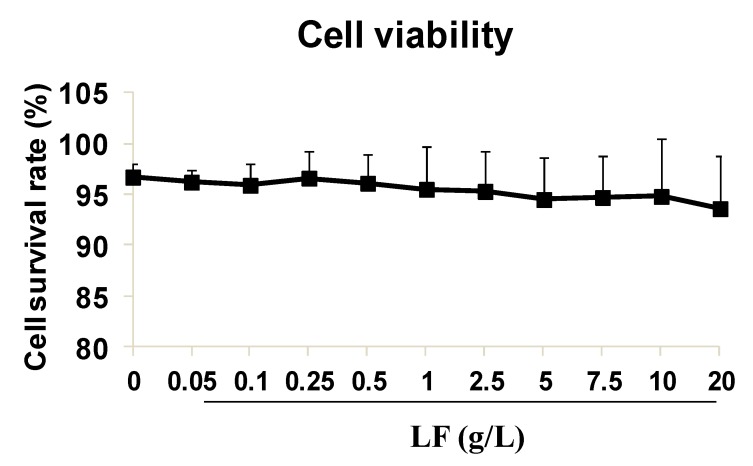
Human umbilical vein endothelial cells (HUVECs) viability detection by cell counting kit-8 (CCK8). Lactoferrin (LF) showed no obvious inhibition on HUVECs survival rate. The data is represen ted as mean ± standard deviation (SD), *n* = 8.

**Figure 2 ijms-20-00587-f002:**
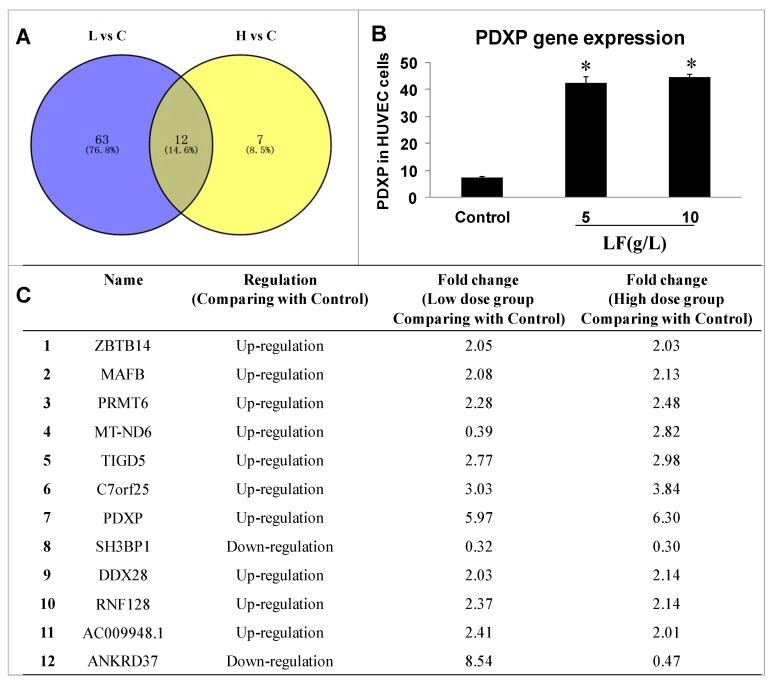
Transcriptomics detection and expression of PDXP. (**A**) Overlapping of selected genes with changed expressions in the control, 5 g/L group (L) and the 10 g/L group (H) through cell transcriptomics detection (*n* = 3). (**B**) Gene expression of PDXP. The data is represented as mean ± SD, * *p* < 0.05, compared with the control (*n* = 3). (**C**) Information about the 12 overlapping genes, including gene name, the regulation condition, and the relative fold change compared with the control group.

**Figure 3 ijms-20-00587-f003:**
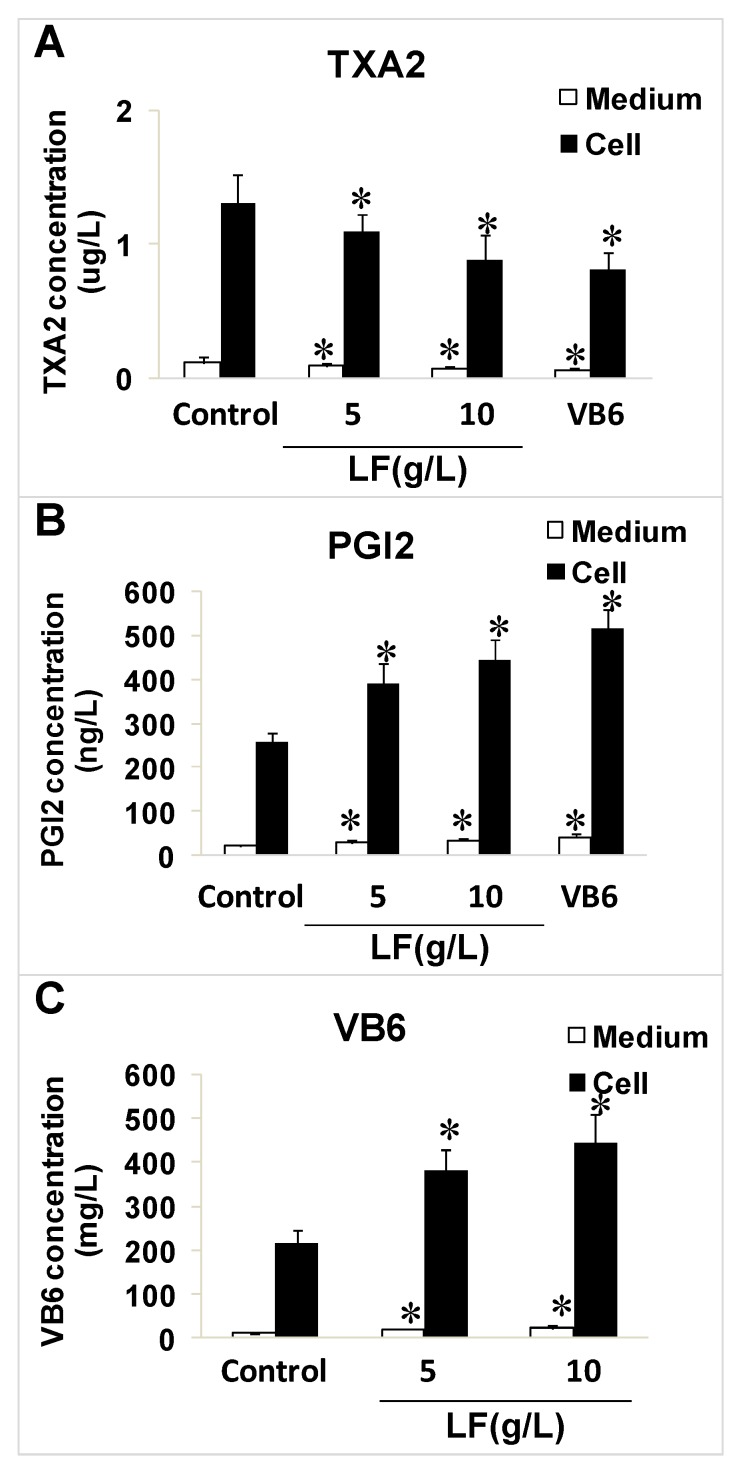
The effect of lactoferrin and VB6 on the levels of TXA2, PGI2. (**A**) TXA2 in cells and medium. (**B**) PGI_2_ in cells and medium. (**C**) VB6 in cells and medium. The final concentration of VB6 was 10 μM. All data are represented as mean ± SD, * *p* < 0.05, compared with the control (*n* = 3).

**Figure 4 ijms-20-00587-f004:**
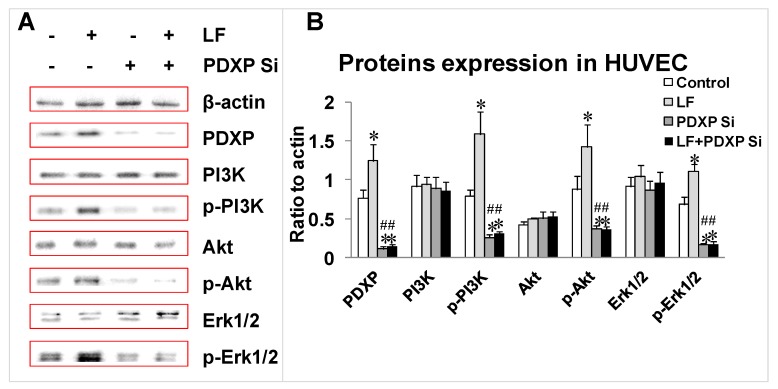
Proteins expression of PDXP, (p-)PI3K, (p-)Akt, and (p-)Erk1/2 treated with of PDXP SiRNA and lactoferrin. (**A**) The protein bands in western blotting detection. (**B**) Statistical analysis of the protein bands. All the data are represented as mean ± SD, * *p* < 0.05, compared with the control. # *p* < 0.05, compared with the PDXP SiRNA group (*n* = 3). Si stands for SiRNA.

**Figure 5 ijms-20-00587-f005:**
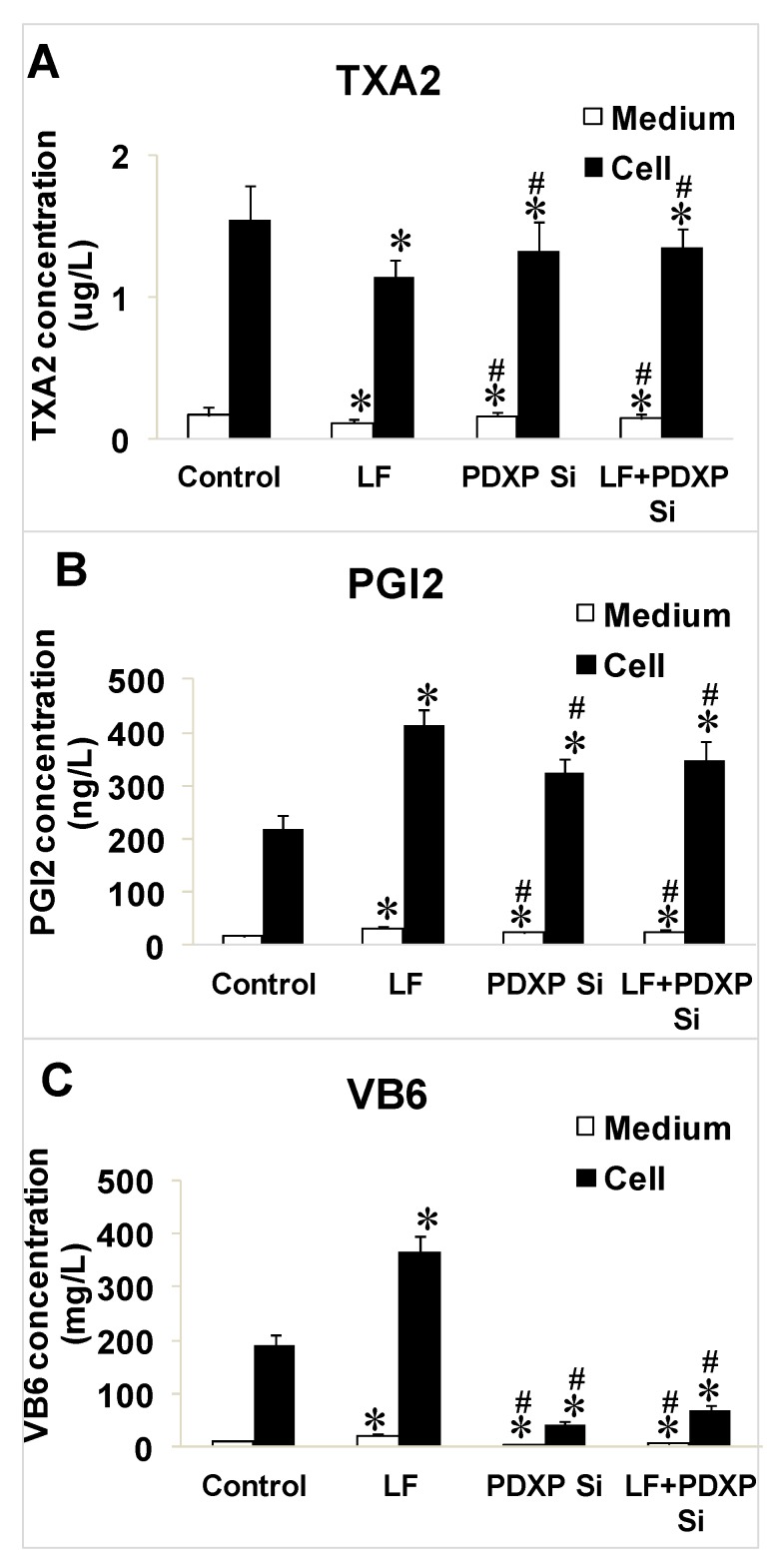
The effect of lactoferrin and PDXP SiRNA on the levels of TXA2, PGI2 and VB6. (**A**) TXA_2_ in cells and medium. (**B**) PGI_2_ in cells and medium. (**C**) VB6 in cells and medium. All the data are represented as mean ± SD, * *p* < 0.05, compared with the control. # *p* < 0.05, compared with the lactoferrin treatment group (*n* = 3). Si stands for SiRNA.

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
