# Peer review of "Lactoferrin Induces the Synthesis of Vitamin B6 and Protects HUVEC Functions by Activating PDXP and the PI3K/AKT/ERK1/2 Pathway"

_ijms, 2019, doi:10.3390/ijms20030587_

Round 1
Reviewer 1 Report
REVIEWER REPORTS:
In this manuscript, the biological activity of lactoferrin in HUVECs was confirmed by two parts. The first part was to screen out the PDXP gene by transcriptomic analysis, which was significantly interfered by lactoferrin. The result demonstrated that lactoferrin could regulate the expressions of VB6, TXA2, and PGI2 by activating PDXP. The second part was to demonstrate lactoferrin could promote cell division by activating the PI3K/AKT/ERK1/2 pathway. The article is well organized, but there were also some pitfalls, some suggestions and comments were presented as follows:
1. The purpose of the current research hasn’t been well described in the Introduction section. Specifically, why did the authors detect PI3K/AKT/ERK1/2 pathway? And what is the relationship among VB6, TXA2 and PGI2? The authors should cite the important relevant references: Recent development of lactoferrin-based vehicles for the delivery of bioactive compounds: Complexes, emulsions, and nanoparticles. Trends in Food Science & Technology, 2018, 79, 67-77.
2. In the discussion part “On the one hand, the PI3K/AKT/ERK1/2 pathway acts as a positive feedback signal in protecting vascular endothelial cells from endogenous damage, because it promotes their proliferation and migration and induces the synthesis of VB6 to maintain their functions (as demonstrated in this study).”, please give more specific explanation and references on the relationship between VB6 and PI3K/AKT/ERK1/2 pathway. And please give some discussion on Fig.4B, why chose PDXP SiRNA to intervene HUVEC?
3. The picture of western blot assay in Figure 4A was unclear and should be improved. Particularly, mosaic could be found in PDXP bound.
4. Supplementary Table 1 showed the “expression of 16 genes decreased in the 10 g/L group relative to the control group (R2)”, but I can’t find PDXP.
Author Response
To reviewer 1:
In this manuscript, the biological activity of lactoferrin in HUVECs was confirmed by two parts. The first part was to screen out the PDXP gene by transcriptomic analysis, which was significantly interfered by lactoferrin. The result demonstrated that lactoferrin could regulate the expressions of VB6, TXA2, and PGI2 by activating PDXP. The second part was to demonstrate lactoferrin could promote cell division by activating the PI3K/AKT/ERK1/2 pathway. The article is well organized, but there were also some pitfalls, some suggestions and comments were presented as follows:
1. The purpose of the current research hasn’t been well described in the Introduction section. Specifically, why did the authors detect PI3K/AKT/ERK1/2 pathway? And what is the relationship among VB6, TXA2 and PGI2? The authors should cite the important relevant references: Recent development of lactoferrin-based vehicles for the delivery of bioactive compounds: Complexes, emulsions, and nanoparticles. Trends in Food Science & Technology, 2018, 79, 67-77.
Answer: We have added some information about the purpose of the current research, as well as the potential relationship among PDXP, VB6 and TXA2/PGI2 (Introduction part, paragraph 2). We have cited the important relevant references in the Introduction part (paragraph 1).
2. In the discussion part “On the one hand, the PI3K/AKT/ERK1/2 pathway acts as a positive feedback signal in protecting vascular endothelial cells from endogenous damage, because it promotes their proliferation and migration and induces the synthesis of VB6 to maintain their functions (as demonstrated in this study).”, please give more specific explanation and references on the relationship between VB6 and PI3K/AKT/ERK1/2 pathway. And please give some discussion on Fig.4B, why chose PDXP SiRNA to intervene HUVEC?
Answer: For the reference about the relationship between VB6 and PI3K/AKT/ERK1/2 pathway is rare, so we added some explanation of the possible relationship between VB6 and PI3K/AKT/ERK1/2 pathway in our study (Discussion part, paragraph 4).
We have added some information to explain why we chose PDXP SiRNA to intervene HUVEC in Discussion part (paragraph 4), to make the manuscript more comprehensive.
3. The picture of western blot assay in Figure 4A was unclear and should be improved. Particularly, mosaic could be found in PDXP bound.
Answer: We have reselected several pictures of western blot assay, to promise the higher resolution of Figure 4A.
4. Supplementary Table 1 showed the “expression of 16 genes decreased in the 10 g/L group relative to the control group (R2)”, but I can’t find PDXP.
Answer: We have carefully checked the data and revised some mistakes. Firstly, in “2.2. Transcriptomic analysis of HUVECs and data analysis” part, we have redrawn the VENN diagram and revised the genes overlapping information to the correct ones; secondly, we added the missing PDXP gene in R2 region and R4 region.
The revised description was stated as follows: “Based on the p value (< 0.05) and the absolute fold change (> 2.0), the expression levels of 75 genes and 19 genes were significantly changed respectively in low dosage group and high dosage group (Supplementary excel), when compared to control group, in which 12 genes were overlapped (Venn diagram in Figure 2A). Among 12 genes, 10 genes were up-regulated and 2 genes were down-regulated comparing with the control, and PDXP was the most significant one with 6-fold change in these 12 genes (Supplementary excel).”
In conclusion, we really appreciate your suggestions and consideration. Thank you very much!

Reviewer 2 Report
The manuscript entitled „Lactoferrin induces the synthesis of vitamin B6 and protects HUVEC functions by activating PDXP and the PI3K/AKT/ERK1/2 pathway” presents the results of studies on the effects of lactoferrin on the biological activity of HUVECs. In my opinion the work is very interesting and innovative. For special attention deserve various methods, that were used, including cell culture and viability testing, siRNA treatment of cells, TXA2, PGI2, and VB6 ELISA detection. Therefore the study presents very comprehensive analysis. The language is fine and the results are generally well presented. In my opinion, the manuscript should be considered for publication. However I have two comments/suggestions: 1) Please change the scale of Y axis in Fig.1 Error bars, scale. I suggest to add error bars. 2) Please provide conclussion section.
Author Response
To reviewer 2:
The manuscript entitled „Lactoferrin induces the synthesis of vitamin B6 and protects HUVEC functions by activating PDXP and the PI3K/AKT/ERK1/2 pathway” presents the results of studies on the effects of lactoferrin on the biological activity of HUVECs. In my opinion the work is very interesting and innovative. For special attention deserve various methods, that were used, including cell culture and viability testing, siRNA treatment of cells, TXA2, PGI2, and VB6 ELISA detection. Therefore the study presents very comprehensive analysis. The language is fine and the results are generally well presented. In my opinion, the manuscript should be considered for publication. However I have two comments/suggestions:
1. Please change the scale of Y axis in Fig. 1 Error bars, scale. I suggest to add error bars.
Answer: We have changed the scale of Y axis in Fig. 1, and added the error bars, to make the figure more clear.
2. Please provide conclusion section.
Answer: We have added some information in conclusion section, to make the sentence more comprehensive.
In conclusion, we really appreciate your suggestions and consideration. Thank you very much!
Reviewer 3 Report
The manuscript by Li et al. reported that transcriptomic analyses indicated that HUVEC cells treated with lactoferrin induce the changes in several gene expressions. The authors selected PDXP as a target for further functional validation experiments. However, the manuscript is very vague, and the research design and rationale are confusing. There are many issues in the data interpretation, and the data are not technical sound to support major conclusions.
Major Comments:
1) For transcriptomic analysis, all genes detected should be reported and appended in the supporting data in order to ensure scientific transparency. It is rarely seen that only less than 100 genes were detected. What method is used to analyze the gene for up/down regulations? What’s the statistical analysis? All are lack of explanation in the manuscript.
2) On page 3, first paragraph, groups R2 and R4 are indicated as decreased expression, but later the authors stated that PDXP was identified as increasing in expression in both R2 and R4 after treatment with lactoferrin. Hard to understand the logic.
3) Why did the authors only choose PDXP instead of other candidate genes for the studies? Did the authors detect changes of TXA2, PGI2, and VB6 related genes in transcriptomics?
4) Why to chose two doses of lactoferrin in the experiments? What’s the purpose for comparison of high/low doses?
5) In figure 1, the data points need error bars.
6) In figure 4, the quality of the Western blots is so low and not legible.
7) What’s the mechanistic relationship between lactoferrin and PDXP, TXA2, PGI2, and VB6? The authors initially stated that they intend to understand the role of lactoferrin in its protection of cardiovascular system, but in the end focus on the effects of PDXP and VB6. A comprehensive discussion on this issue is missing.
8) In the method section, 4.3. Transcriptomic and data analyses, ref [34] is missing.
Author Response
To reviewer 3:
The manuscript by Li et al. reported that transcriptomic analyses indicated that HUVEC cells treated with lactoferrin induce the changes in several gene expressions. The authors selected PDXP as a target for further functional validation experiments. However, the manuscript is very vague, and the research design and rationale are confusing. There are many issues in the data interpretation, and the data are not technical sound to support major conclusions.
Major Comments:
1. For transcriptomic analysis, all genes detected should be reported and appended in the supporting data in order to ensure scientific transparency. It is rarely seen that only less than 100 genes were detected. What method is used to analyze the gene for up/down regulations? What’s the statistical analysis? All are lack of explanation in the manuscript.
Answer: Firstly, we have added the information of analysis method in Method part, to make the study more clear and comprehensive.
Secondly, we have checked the data carefully and redrawn the VENN diagram (in “2.2. Transcriptomic analysis of HUVECs and data analysis” part), then revised the genes overlapping information to the correct one.
Thirdly, we have uploaded data excel containing all the genes as the supplementary material, to promise the scientific transparency.
2. On page 3, first paragraph, groups R2 and R4 are indicated as decreased expression, but later the authors stated that PDXP was identified as increasing in expression in both R2 and R4 after treatment with lactoferrin. Hard to understand the logic.
Answer: We really feel sorry for the confusing information we primarily provided. We have checked the data and genes overlapping information carefully in Supplementary excel, and redrawn VENN diagram. R2 and R4 was proved to be groups of “High dosage group higher than control” and “Low dosage group higher than control”, respectively, which further validated that PDXP was identified to be up-regulated (6-fold increase vs control) with the treatment of lactoferrin.
3. Why did the authors only choose PDXP instead of other candidate genes for the studies? Did the authors detect changes of TXA2, PGI2, and VB6 related genes in transcriptomics?
Answer: We have screened out several candidate genes in transcriptomics detection, and PDXP was the most significant one with 6-fold change, therefore, PDXP was selected to investigate the related mechanism in the present study. The other candidate genes and their related pathways will be investigated in the other researches. The genes related with TXA2, PGI2 and VB6 had not been detected in this research, however, we will pay more attention to them in another study recently. Thank you very much for your meaningful suggestions.
4. Why to chose two doses of lactoferrin in the experiments? What’s the purpose for comparison of high/low doses?
Answer: Primarily, the two doses of lactoferrin (5g/L and 10g/L) were chosen to select candidate genes through comparing their expression levels, that is, the gene level showing tendencies of “control>low dosage group>high dosage group” or “control<low dosage group<high dosage group” was chosen as the valid one in data analysis part. On the other hand, the two doses was chosen further to investigate the dosage-effect relationship.
5. In figure 1, the data points need error bars.
Answer: We have added the scale and error bars in Y axis in Figure 1, to make it more clear.
6. In figure 4, the quality of the Western blots is so low and not legible.
Answer: We have reselected other western blots with high resolution in Figure 4, and calculated the expression level by software.
7. What’s the mechanistic relationship between lactoferrin and PDXP, TXA2, PGI2, and VB6? The authors initially stated that they intend to understand the role of lactoferrin in its protection of cardiovascular system, but in the end focus on the effects of PDXP and VB6. A comprehensive discussion on this issue is missing.
Answer: The mechanistic relationship between lactoferrin and PDXP, TXA2, PGI2, and VB6 can be summarized as follows: in our study, we found that lactoferrin could increase the synthesis of VB6 through activating PDXP gene, and further down-regulate TXA2 and up-regulate PGI2, in HUVECs.
To make the manuscript more clear and comprehensive, we have added the explanation about the protection of lactoferrin on cardiovascular system in Discussion part.
8. In the method section, 4.3. Transcriptomic and data analyses, ref [34] is missing.
Answer: We have checked the manuscript and references carefully, and found that ref [34] was indeed missing, which was tightly related with the data analyses in transcriptomic detection. Therefore, we added the cited ref [34] in 4.3. section.
In conclusion, we really appreciate your suggestions and consideration. Thank you very much!

Round 2
Reviewer 3 Report
The authors have made major revision to address reviewers’ concerns and improved the manuscript. It is appreciated that the authors uploaded the full transcriptomics data as supportive info to enhance the scientific transparency. However, additional revisions are still necessary to improve the clarity of the data presentation in the manuscript.
Major concerns:
1. For Figure 2, please add a table to tabulate the overlapping 12 genes in the VENN diagram, what they are, whether is up or down regulation compared to the controls, and how many fold changes they are. Please make necessary self-description in the figure caption that readers are easy to follow the figure. Since this info is the key and rationale for the current study, you will not expect readers to check the supportive data to pick those genes of interest.
2. The authors answer the question 3. Why did the authors only choose PDXP instead of other candidate genes for the studies? Did the authors detect changes of TXA2, PGI2, and VB6 related genes in transcriptomics? “Answer: We have screened out several candidate genes in transcriptomics detection, and PDXP was the most significant one with 6-fold change, therefore, PDXP was selected to investigate the related mechanism in the present study. The other candidate genes and their related pathways will be investigated in the other researches. The genes related with TXA2, PGI2 and VB6 had not been detected in this research, however, we will pay more attention to them in another study recently.”
Please incorporate this answer in a proper way into the manuscript to enhance the clarity of the data presentation.
Author Response
Major concerns:
1. For Figure 2, please add a table to tabulate the overlapping 12 genes in the VENN diagram, what they are, whether is up or down regulation compared to the controls, and how many fold changes they are. Please make necessary self-description in the figure caption that readers are easy to follow the figure. Since this info is the key and rationale for the current study, you will not expect readers to check the supportive data to pick those genes of interest.
Answer: We have added a table to tabulate information about the overlapping 12 genes (in Figure 2C), including gene name, regulation condition and fold changes. Meanwhile, we have added the description about the 12 genes in figure caption, to make the figure more comprehensive.
2. The authors answer the question 3. Why did the authors only choose PDXP instead of other candidate genes for the studies? Did the authors detect changes of TXA2, PGI2, and VB6 related genes in transcriptomics? “Answer: We have screened out several candidate genes in transcriptomics detection, and PDXP was the most significant one with 6-fold change, therefore, PDXP was selected to investigate the related mechanism in the present study. The other candidate genes and their related pathways will be investigated in the other researches. The genes related with TXA2, PGI2 and VB6 had not been detected in this research, however, we will pay more attention to them in another study recently.”
Please incorporate this answer in a proper way into the manuscript to enhance the clarity of the data presentation.
Answer: We have incorporated these sentences into the manuscript (in Result part and Discussion part), to promise the expression to be more clear and comprehensive.
In conclusion, we really appreciate your suggestions and consideration. Thank you very much!
